# Research on High-Temperature Evaluation Indexes and Performance of Qingchuan Rock-SBS Composite Modified Asphalt

**DOI:** 10.3390/ma15217455

**Published:** 2022-10-24

**Authors:** Yan Hao, Yali Ye, Chuanyi Zhuang, Fengjian Hou

**Affiliations:** 1School of Transportation and Civil Engineering, Shandong Jiaotong University, Jinan 250357, China; 2Donghui Testing & Certification Group Co., Ltd., Jinan 250024, China

**Keywords:** road engineering, Qingchuan rock asphalt, high-temperature evaluation, double-layer composite structure, multiple evaluation indexes

## Abstract

The phenomenon of structural destabilization damage to asphalt pavement is becoming increasingly serious as a result of high temperatures and heavy traffic. Considering the advantages of Qingchuan rock asphalt (QRA) in its durability, high-temperature rutting resistance, and good compatibility with asphalt, it was proposed to compound rock asphalt with SBS to ameliorate the high-temperature performance of asphalt. In this study, DSR and BBR were used to determine the rheological properties of Qingchuan rock-modified asphalt (QRMA) and Qingchuan rock–SBS-modified asphalt (QRA-SBSMA), and the optimum blending amount of rock asphalt was determined based on the PG classification results. Secondly, four different structures of ‘30 mm AC-10 upper layer (70-A, QRMA, SBSMA, QRA-SBSMA) + 50 mm AC-16 lower layer (70-A)’ double-layer composite specimens were prepared. Multiple high-temperature performance evaluation indexes (G*/sinδ, Ds, rutting depth, micro-strain, Fn, modulus) were used to assess the improvement effect of QRA. Finally, using a 1/3 scale accelerated loading testing machine, we simulated high-temperature, water, and high-temperature coupled environments to assess the impact of high temperature and water on the performance of QRMA and QRA-SBSMA, respectively. The findings demonstrated that QRA can increase the PG classification of 70-A and SBSMA as well as its resistance to high-temperature deformation. Multi-index comprehensive evaluation methods were used to consummate the asphalt high-temperature evaluation system. The QRA-SBSMA had the smallest rutting depth and creep rate and the largest dynamic modulus, characterizing its ability to optimally resist high-temperature rutting and deformation.

## 1. Introduction

Along with the rapid development of the transportation industry, the traffic volume has been growing rapidly. The ratio of overloaded vehicles is large, which leads to serious ruts and other diseases at highway intersections and uphill sections, and seriously affects the comfort and safety of driving [1,2,3,4]. Qingchuan rock asphalt (QRA) is not only rich in reserves but also has the advantages of high nitrogen content, good durability, and good compatibility with base asphalt. The preparation process is simple and commonly used in asphalt modification [5,6]. Rock asphalt can not only greatly improve the stability and shear strength of asphalt pavement under high temperature and reduce rutting deformation but also delay the formation of cracks and extend the service life of pavement [7,8]. Rock asphalt mainly includes North American rock, Burton rock, Qingchuan rock, and Xinjiang rock. The performance of modified asphalt varies depending on the type of rock asphalt used [9,10,11].

Su [12] described how, while rock asphalt and base asphalt have good compatibility, segregation is not easy to achieve in modified asphalt. Yan [13] proposed that the mixing ratio of rock asphalt should be optimized based on the specific performance. Chen [14] optimized the preparation technology of Qingchuan rock modified asphalt and reported that the high-temperature deformation resistance of base asphalt was improved after reasonable shearing and development time. Li [15] summarized the test methods for evaluating the high-temperature performance of rock asphalt and described their use of DSR to measure the rheological properties of asphalt; the high-temperature performance evaluation was more accurate. In the evaluation of asphalt rheological properties, Lv [16] and Wen [17] tested the rheological properties of Buton rock asphalt before and after aging. Through temperature scanning and multiple stress creep recovery tests, it was found that Buton rock asphalt can not only enhance the high-temperature performance of base asphalt. Asphalt made by mixing rock asphalt with other modifiers performed better than that of a single modifier. Cheng [18] and Zhong [19] proposed using Xinjiang rock asphalt to increase the high-temperature PG classification. The high-temperature performance was improved but also had good tensile performance and fatigue performance. Li [20] used the frequency–temperature scanning test and creep relaxation test of bending beams and found that Buton, Qingchuan, and North American rock asphalt enhanced the stiffness of the material and slightly reduced the relaxation stress at low temperatures. Teltayev [21] pointed out that in the BBR test, E < 300 MPa was more stringent and more suitable for evaluating low-temperature performance than m > 0.3. In terms of microstructure, Wang [22] and Zhang [23] observed the surface structure of Buton rock asphalt with the help of AFM and SEM. The results revealed that the rock asphalt surface was smooth and coarse without honeycomb phase structure. FT-IR measurements of Ma [24] showed that there was no other chemical between natural rock asphalt and base asphalt. 

In terms of the evaluation of asphalt mixture, Zou [25] reported that asphalt binder had a significant effect on the rutting resistance of asphalt mixture. Mixtures modified by Qingchuan rock can reduce the rutting depth and improve the high-temperature stability to a large extent. Guo [26] explored the factors influencing high-temperature rutting resistance and showed that the type of asphalt was the main reason affecting the structural properties of asphalt mixtures, followed by the environment. Liu [27] conducted a mix ratio design of rock-asphalt-modified Superpave mixture and pointed out that the rutting test and dynamic modulus test could be used to evaluate high-temperature performance. Zhang [28] modified Qingchuan rock asphalt and rubber powder and designed a single-layer AC-13 mixture. According to the high-temperature rutting test, low-temperature beam test, and water stability, rock asphalt can complement with SBR asphalt in high and low temperature performance [29,30], such that the performance of rock asphalt and rubber powder was superior to SBS-modified asphalt. Ren [31] characterized and predicted the rutting depth change of rock asphalt mixture utilizing an improved rutting test, indicating that rock asphalt had excellent water damage resistance and could enhance the high-temperature performance of base asphalt. Li [32] made a mixture of Qingchuan rock asphalt from northern Sichuan province and North American rock asphalt. Based on the results of high-temperature dynamic stability, freeze-thaw splitting strength, and indirect tensile fatigue tests, it was noted that Qingchuan rock asphalt was better when modified and could not only resist low-temperature environments as low as −21.5 °C, but also had better high-temperature and fatigue performance than North American rock asphalt. Li [33] studied the modification of various types of rock asphalt and reported that the chemical composition of rock asphalt had a significant impact on pavement performance. North American, Qingchuan, and Xinjiang rock asphalt were compatible with base asphalt, and Qingchuan rock had a positive influence on the high-temperature performance, fatigue performance, and durability of the mixture, while the effect on low-temperature performance was different at different dosages. It was also proposed that the traditional asphalt evaluation index was not fully applicable to the evaluation of the rock asphalt modification effect. Li [34] and Zhang [35] studied the self-healing performance and fatigue characteristics of rock asphalt. The optimal self-healing temperature of rock asphalt modified asphalt was lower than the phase transition temperature of the Newtonian liquid, which had good self-healing potential. It could alleviate the time of fatigue cracks in the mixture and improve fatigue performance.

At present, the research on rock asphalt mainly focuses on Buton rock, Qingchuan rock, Buton–SBR composite, or Buton with another modifier composite, with less research on Qingchuan rock asphalt with SBS-composite-modified asphalt. Simultaneously, the high-temperature evaluation means used in the mixture is relatively single, mostly are wheel-tracking testing and Hamburg wheel-track testing. Due to the limitation of loading times of both, the test results do not cover the whole results of the asphalt layer rutting damage. The expression of rutting deformation law is not strict enough, which leads to incomplete evaluation of asphalt high-temperature performance. Therefore, this study characterized the high-temperature property of QRA-SBSMA through a variety of evaluation methods and made a more comprehensive supplement to the high-temperature evaluation system, including multiple creep testing and dynamic modulus testing. In particular, the accelerated loading equipment is used to study the rutting failure law of the asphalt layer by setting different loading environments and times. This paper improves the high-temperature evaluation system of asphalt mixture, not only for rock asphalt, but also provides more valuable reference for the high-temperature performance evaluation methods and practical application of other asphalt modifiers.

## 2. Materials and Methods

### 2.1. Materials

Type 70-A base asphalt was used in the test. According to the test methods in JTG E20-2011 ‘Standard Test Methods of Bitumen and Bituminous Mixtures for Highway Engineering’ [35], the performance indexes obtained are shown in Table 1. The technical indexes of Qingchuan rock asphalt, SBS modified asphalt, and filler are shown in Table 2, Table 3 and Table 4.

All the above materials met the requirements of JTG F40-2004 ‘Technical Specifications for Construction of Highway Asphalt Pavements’ [36].

### 2.2. Rheological Property Testing Methods

The rutting factor, creep stiffness, and *m*-value of different Qingchuan rock contents were measured by DSR (Bohlin Gemini 200 ADS, Worcestershire, Malvern Instruments, Worcestershire, UK) and BBR (TE-BBR-F, Cannon Instruments, State College, PA, USA). Through the PG grading results of rock modified asphalt and composite modified asphalt, the optimum blending amount of Qingchuan rock was determined.

According to the requirements of AASHTO T315 [37], DSR temperature was 64~88 °C, each test interval was 6 °C. The sinusoidal load was applied using the strain control mode. The specimen diameter was 25 mm, the thickness was 1 mm, the target strain value was 12%, and the loading frequency was 10 rad/s. The measured rutting factor is the ratio of the complex modulus to the sine of the phase angle, and a larger value also represents better high-temperature rutting resistance.

According to the requirements of AASHTO TP125 [38], the temperature of BBR was −6~−18 °C, the sample size was 125 mm × 6.25 mm × 12.5 mm. Three points of bending loading were used to apply a vertical load of 980 mN ± 50 mN. The creep stiffness *s* and *m*-value at 60 s were recorded.

The content of Qingchuan rock in modified asphalt was 3%, 5%, 7%, and 9% respectively. The content of Qingchuan rock in composite modified asphalt was 2.5%, 5%, and 7.5%, respectively. Prepare rock modified asphalt and composite modified asphalt according to the following steps:

(1) Qingchuan rock-modified asphalt (QRMA)

The base asphalt was placed in the oven at 140 °C. When it could be stirred evenly, it was placed on the shear, and the rock asphalt powder was added in small amounts and several times in 20 min at a speed of 1500 r/min. Subsequently, the stirring temperature was adjusted to 170 °C, the stirring speed was 3000 r/min and the stirring time was 30 min. Finally, the asphalt was placed in the oven at 170 °C for 40 min development to produce rock modified asphalt. Its performance index test results are shown in Table 5.

(2) Qingchuan rock asphalt and SBS-composite-modified asphalt (QRA-SBSMA)

The finished SBS modified asphalt was baked to 180 °C in the oven, and then the rock asphalt powder was added in small amounts and several times in 20 min on the shear meter at a speed of 1500 r/min. Then, the stirring speed was adjusted to 3000 r/min shear for 30 min. Finally, the asphalt was placed in an oven at 180 °C for 40 min to prepare Qingchuan rock–SBS-composite-modified asphalt. The performance index test results are shown in Table 6.

Qingchuan rock asphalt contains more colloid and asphaltene. Along with the incorporation of rock asphalt, the content of hard components in the modified asphalt increased continuously, which made the asphalt colloid move towards the gel direction, the viscosity increased, and the macroscopic performance showed reduced penetration. The incorporation of rock asphalt increased the asphaltene component of the modified asphalt, enhanced the polar bonds, and increased the softening point. The enhanced intermolecular forces of asphalt also led to a reduction in the ability of asphalt molecules to produce slip, reducing the low-temperature ductility value.

Through high-speed shearing, the dispersion uniformity of rock asphalt in SBS modified asphalt was observed by Carl Zeiss Axiocam 506 color fluorescence microscope. Blue light was used for excitation observation and the fluorescence microscope was magnified at 100 and 200 times.

From the Figure 1 of the fluorescence microscope results, the ash in rock asphalt did not have a fluorescent state. Qingchuan rock asphalt could be uniformly dispersed in SBS modified asphalt, with good compatibility, and no agglomeration. Qingchuan rock could be blended with SBS asphalt to form a cross-linked network with good dispersion and modification effect.

### 2.3. High-Temperature Performance Evaluation of Indoor Test

The mixture structure studied in this experiment was a composite structure of ‘lower layer AC-16 + upper layer AC-10’, which used 10–16 mm, 5–10 mm, 3–5 mm, 0–3 mm four-grade gravel. The design gradation is shown in Table 7, and the design scheme of the double-layer composite structure specimen is shown in Table 8.

The low-temperature index and fatigue index of the four groups of double-layer composite structures met the requirements of JTG F40-2004 ‘Technical Specifications for Construction of Highway Asphalt Pavements’ [36]. Additionally, this study focused on the various methods to evaluate the high-temperature performance of QRMA and QRA-SBAMA; the low-temperature performance and fatigue performance results are no longer listed in detail.

#### 2.3.1. Wheel-Tracking Testing

According to the technical requirements of ‘Standard Test Methods of Bitumen and Bituminous Mixtures for Highway Engineering’ [35] (JTG E20-2011) T0719, the specimen adopted a 50 mm AC-16 + 30 mm AC-10 double-layer structure. The test temperature was 60 °C, the wheel pressure was 0.7 MPa, and the rutting test size was 300 mm × 300 mm × 80 mm.

#### 2.3.2. Hamburg Wheel-Track Testing

Hamburg wheel-track testing can simulate the most adverse conditions suffered by real road surfaces, while evaluating high-temperature performance and water damage resistance. According to the AASHTO T324-04 standard [39], the water bath temperature was 50 °C, and the specimen size was 150 mm × 60 mm. The deformation was measured by rolling 20,000 times with 0.73 MPa wheel pressure.

#### 2.3.3. Multiple Creep Testing

The test procedure refers to the multiple creep testing in the U.S. code and the repeated loading permanent deformation test recommended by the CHRP (National Cooperative Highway Research Program). The test was carried out under the conditions of 30 °C, 45 °C, and 60 °C respectively under the condition of 700 kPa compressive stress without confining pressure. The size of the specimen was *φ* 100 mm × 150 mm, and the repeated axial half-sine load was applied to the specimen. The loading frequency was 0.1 s per second and the interval was 0.9 s. The termination condition was that the number of repeated loadings had reached 20,000 times or the structure had reached 5000 micro-strains.

#### 2.3.4. Dynamic Modulus Testing

According to the ‘Standard Test Methods of Bitumen and Bituminous Mixtures for Highway Engineering’ [35] (JTG E20-2011) T0378, the temperature setting range was −10~60 °C. The AMPT testing machine was used to test the dynamic modulus of asphalt mixtures with different structural types without confining pressure. Different loading frequencies (25 Hz, 20 Hz, 10 Hz, 5 Hz, 2 Hz, 1 Hz, 0.5 Hz, 0.2 Hz, 0.1 Hz) and different test temperatures (20 °C, 35 °C, 45 °C, 50 °C) were selected to analyze the law of dynamic modulus variation of mixture with loading frequency and test temperature.

### 2.4. High-Temperature Performance Evaluation of Accelerated Loading Testing

Hamburg wheel-track testing was limited by the number of loading times and cannot analyze the asphalt layer covering the whole failure process. In order to better simulate the variation characteristics of the actual pavement under different traffic loads, the 1/3 scale accelerated loading device was adopted in this paper, as shown in Figure 2. By adjusting the axial load and temperature, the destructive effect of different wheel loads on structural specimens was simulated. The performance index variation law of the double-layer pavement structure was analyzed to evaluate the long-term performance of the asphalt pavement structure.

#### 2.4.1. High-Temperature Environment

According to JTG E20-2011 ‘Standard Test Methods of Bitumen and Bituminous Mixtures for Highway Engineering’ [35], 80 mm thick rutting specimens were prepared; 60 °C was used as the control temperature, and the wheel load was 0.7 MPa. Under the action of moving tire load, the rolling was terminated when the rutting depth exceeded 10 mm. The rutting depth of the specimen was collected every 3600 times (1 h).

#### 2.4.2. Water and High-Temperature Coupling Environment

When the double-layer composite specimen was measured for coupling effect in water and high-temperature environment, the specimen preparation and wheel pressure setting were the same as those in Section 2.4.1. Accelerated loading testing was performed in a water bath by filling the test tank with water at 50 °C so that the liquid level was flush with the top of the sample. The number of rolling tests was set to 30,000 times, and the depth of rutting was measured every 5000 times.

## 3. Results and Discussion

### 3.1. Rheological Testing Results of Modified Asphalt

#### 3.1.1. Dynamic Shear Rheometer Testing Results

The rheological characteristics of DSR were used to judge the high-temperature performance of different Qingchuan rock contents. The rutting factor results of 64~88 °C are shown in Figure 3.

According to the protocol in Table 9 and Table 10, it can be seen in the subsequent tests: at the same temperature, the rutting factor tended to increase with the increasing rock asphalt content. Qingchuan rock asphalt is a solid petroleum-based material with a chemical structure similar to that of asphalt and has excellent compatibility with base asphalt. Rock asphalt improved the asphaltene content in the base asphalt, increased the stiffness of bitumen, and the effect of high-temperature rutting was greatly improved.

When the temperature rose, the sensitivity of asphalt molecular chain motion increased, which made the deformation resistance of rock asphalt modified asphalt decrease. The G*/sinδ of 7% and 9% rock asphalt reached more than 1 kPa at 76 °C. The high-temperature grade was 76 °C, which was much larger than base asphalt. This showed that rock asphalt could significantly enhance the high-temperature performance of base asphalt and improve the anti-rutting ability.

Rock asphalt was ground and dispersed at high speed by shears, which can interweave with SBS-modified asphalt to form a mutually cross-linked mesh structure and enhance the rutting factor. Rock asphalt could not only improve the intermolecular force and enhance the polar bond but also improve the high-temperature performance of SBS-modified asphalt.

#### 3.1.2. Bending Beam Rheometer Testing Results

To study the influence of rock asphalt on the low-temperature performance of base asphalt and SBS-modified asphalt, the S and *m*-value of 70-A, QRMA, and QRA-SBSMA were measured by BBR test. The test results are shown in Figure 4 and Figure 5.

The flexibility decreased and brittleness increased of modified asphalt after the addition of rock asphalt. The creep stiffness of asphalt with rock asphalt content increased, and the *m*-value showed the opposite trend. This indicated that the incorporation of rock asphalt weakened the low-temperature deformation resistance and performance of asphalt. Therefore, the content of rock asphalt should not be too high, otherwise, it will affect the low-temperature performance of QRMA and QRA-SBSMA.

For PG classification results of Table 11 and Table 12, considered the adverse effects of rock asphalt on high-temperature and low-temperature performance. Finally, the optimum blending amount of rock asphalt in QRMA was 7%, and in QRA-SBSMA, it was 2.5%.

### 3.2. Laboratory Testing Results of Mixtures

#### 3.2.1. Indoor Rutting Testing Results

The indoor rutting testing was carried out on the 80 mm double-layer composite specimen, and the rutting depth deformation results are shown in Figure 6. The dynamic stability results of the four schemes are shown in Table 13.

The four schemes had the same lower layer structure, all of which were base asphalt mixture. The rutting depth and rutting growth rate of the base asphalt mixture were significantly greater than those of the other three mixtures throughout the entire loading period, as shown by the rutting depth. Compared with Scheme 1, Scheme 2 reduced rutting depth by 36.3% and improved dynamic stability by 54.4%. Although rock asphalt was shown to improve the high-temperature performance of base asphalt, it was not as effective as SBS.

QRA-SBSMA rutting depth and dynamic stability were slightly better than SBS-modified asphalt, indicating that incorporating rock asphalt into SBS modified asphalt can help improve high-temperature rutting resistance, but not as effectively as improving the base asphalt.

#### 3.2.2. Hamburg Wheel-Track Testing Results

Hamburg wheel-track testing is widely regarded as one of the most rigorous devices for investigating the high-temperature and water-damaged properties of asphalt mixtures. The comparison of deformation with rolling times was shown in Figure 7.

In the coupled environment of water and high-temperature, the wheel load action had the greatest effect on the base asphalt mixture, with deformation of 15 mm for only 15,000 loads. The other three modified asphalt rutting deformation trends were less different. However, because of the high nitrogen content of QRA, the macroscopic performance of the mixture was strong resistance to water damage, and the QRA-SBSMA rutting depth was the smallest.

While the rock asphalt was compatible with the base asphalt, the large molecules in the rock asphalt underwent cleavage at high temperatures and polymerized with the small molecules, resulting in an overall increase in adhesion. Under coupling water and high-temperature conditions, rock asphalt not only improved the high-temperature performance of base asphalt but also outperformed SBS-modified asphalt.

#### 3.2.3. Multiple Creep Testing Results

Multiple creep testing was carried out to study the resistance of various asphalt mixtures to permanent deformation. The flow number (Fn) was defined as the number of loadings corresponding to the minimum change rate of permanent axial strain. The micro-strain and Fn results corresponding to different loading times are shown in Figure 8.

Figure 8a–c showed that the number of repetitive loadings to reach 50,000 micro-strains decreased as the temperature increased. The number of loadings corresponding to 50,000 micro-strains at 60 °C was much smaller than that of 30 °C and 45 °C, indicating that temperature had a significant impact on the deformation capacity of the structure to resist the load action. The test results were Scheme 4 > Scheme 2 > Scheme 3 > Scheme 1, whether it was the 20,000 loading times or 50,000 micro-strains. The deformation resistance of Qingchuan rock asphalt was higher than that of SBS-modified asphalt. The variation pattern of Fn was the same as micro-strain, which showed that adding rock asphalt can greatly enhance the deformation resistance of the mixture under repeated loading at high temperature.

#### 3.2.4. Dynamic Modulus Testing Results

The trends of dynamic modulus with temperature and loading for the four structural schemes are shown in Figure 9.

The dynamic modulus of the four mixture schemes tended to decrease significantly as the temperature or frequency of load action increased, and the rate of decrease gradually slowed until it flattened out. This was due to the asphalt stiffness modulus gradually reducing and a weakening of the resilience under stress, as manifested by a decrease in dynamic modulus. At the same temperature, the dynamic modulus gradually decreased while the rate increased. QRMA mixture had a higher dynamic modulus than base asphalt mixture and SBS modified asphalt mixture, indicating that QRA had a favorable effect on the dynamic modulus of base asphalt. The ability to resist frequency loading was ranked as Scheme 4 > Scheme 2 > Scheme 3 > Scheme 1, with the best performance in QRA-SBSMA.

The factors affecting the performance of asphalt mixtures differ at high temperatures and low frequencies versus low temperatures and high frequencies. The binder in asphalt has a greater impact at low temperatures and high frequency loading. When the temperature rises or the load frequency falls to a certain level, the asphalt binder viscosity decreases, and the impact on the performance of the asphalt mixture decreases. At this point, the mineral skeleton becomes the most important factor influencing the asphalt mixture, and the difference in modulus values between the four mixture schemes is diminutive.

### 3.3. Accelerated Loading Testing Results of Double-Layer Structure

Accelerated loading testing can more realistically simulate the damaging effects of traffic loads on pavement structures. Simulation of different working conditions is achieved by varying the speed and weight of the rolling wheel and controlling the temperature of the loaded pavement. The purpose of the accelerated loading testing study is to compare the rutting deformation resistance of four structure types at the same temperature and number of loads. To better evaluate the road performance of the mixture, a 1/3 scale pavement accelerated loading testing machine was used to analyze different structures covering the entire damage process and to evaluate the whole life cycle.

#### 3.3.1. Accelerated Loading High-Temperature Testing Results

To ensure the accuracy of the accelerated loading testing, three sets of parallel specimens were used for each scheme. The average value of the rutting depth of the parallel specimens was used as the test result under the environment of 60 °C.

As can be seen from Figure 10: the base asphalt mixture had the worst rutting resistance, with a rutting depth of 11 mm for only 18,000 loadings. Both QRA-modified asphalt and SBS-modified asphalt can significantly reduce the rutting depth, improving the rutting resistance ability. The change pattern of the four schemes rutting depth coincided with the conventional rutting testing, and the growth rates were all rapid firstly and then appeared to grow steadily into a stable development stage. The rutting depth tended to increase linearly, but it had not yet entered the accelerated damage phase.

#### 3.3.2. Accelerated Loading High-Temperature and Water Testing Results

In the 50 °C water bath environment, the rutting depth of different schemes varies with the loading times, as shown in Figure 11.

Compared to the Hamburg wheel-track testing, the number of loads in this test was increased by 10,000 times. In the Hamburg wheel-track testing, an increase in the growth rate of rutting depth was observed at 15,000 to 20,000 crushing times. The water bath accelerated loading testing showed essentially the same variation pattern, but the loading conditions were more severe. After the number of loadings exceeded 20,000, the rutting growth rate tended to level off, indicating that the rutting started to enter a stage of steady growth. The resistance to water damage was ranked as follows: Scheme 4 > Scheme 2 > Scheme 3 > Scheme 1. It was proved that QRA can significantly enhance the intermolecular force and chemical cross-linking, improving the water repellency and adhesion of the QRA-SBSMA. It improves the bonding between aggregate and asphalt, making it more resistant to water and high-temperature damage.

## 4. Conclusions

This paper aimed to characterize the performance improvement effect of QRA on SBS modified asphalt through multi-index comprehensive evaluation methods, and consummate the high-temperature evaluation system for mixtures. Through high-temperature tests under different conditions, the following conclusions can be drawn:

Qingchuan rock had different degrees of improvement effect on the high-temperature performance of base asphalt and SBS modified asphalt. For the PG classification results and taking into account the adverse effects of rock asphalt on low-temperature performance, the optimum blending amount of rock asphalt in QRMA and QRA-SBSMA was 7% and 2.5%, respectively.

It was not rigorous enough to characterize the high-temperature performance of the mixture using only the rutting deformation from the wheel-tracking testing and the Hamburg wheel-track testing. In this study, multiple creep testing and dynamic modulus testing were added to measure the rheological number and dynamic modulus, and the obtained test results were ranked differently from the previous two. Combined with accelerated loading tests, 60 °C rutting testing and 50 °C water-bath rutting testing were carried out to analyze the variation pattern of rutting depth. For the accuracy of the high-temperature performance study results, the multi-index comprehensive evaluation methods were required.

Through the multi-index comprehensive evaluation methods, high-temperature performance ranked in a comprehensive order of QRA-SBSMA > SBSMA > QRMA > 70-A. Qingchuan rock- and SBS-composite-modified asphalt has both the hardness of rock asphalt and retains the toughness of SBS-modified asphalt, with more excellent resistance to high-temperatures and water damages.

In this paper, no infrared spectroscopy testing was conducted on QRA-SBSMA, which lacked the investigation of the modification mechanism and should be added subsequently. This paper studied the indoor tests and accelerated loading tests of composite modified asphalt, the next step should be supplemented with field testing and finite element theoretical simulation to proceed with a more in-depth study of the evaluation index.

## Figures and Tables

**Figure 1 materials-15-07455-f001:**
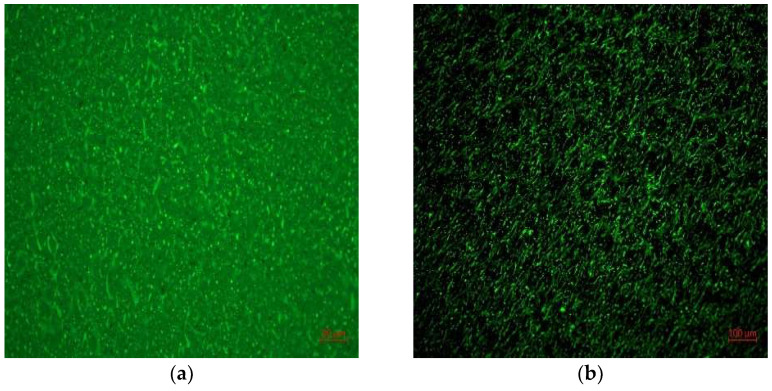
Modified asphalt fluorescence results: (**a**) dispersion in SBSMA; (**b**) crosslinking with SBSMA.

**Figure 2 materials-15-07455-f002:**
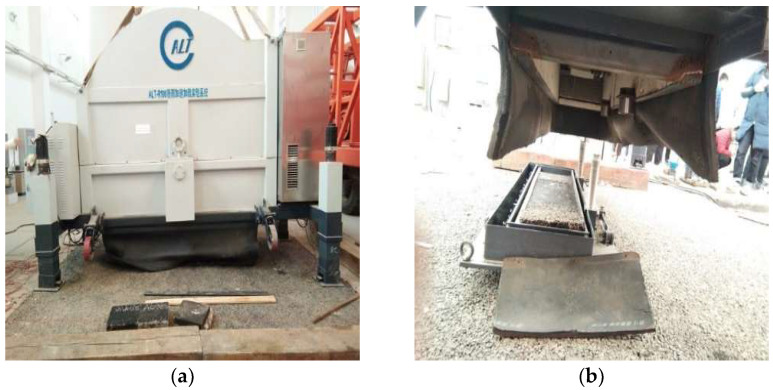
(**a**) Accelerated loading equipment; (**b**) testing tank.

**Figure 3 materials-15-07455-f003:**
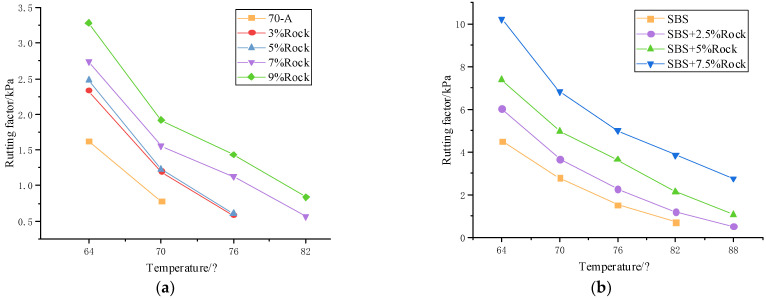
Rutting factor testing results:(**a**) QRMA; (**b**) QRA-SBSMA.

**Figure 4 materials-15-07455-f004:**
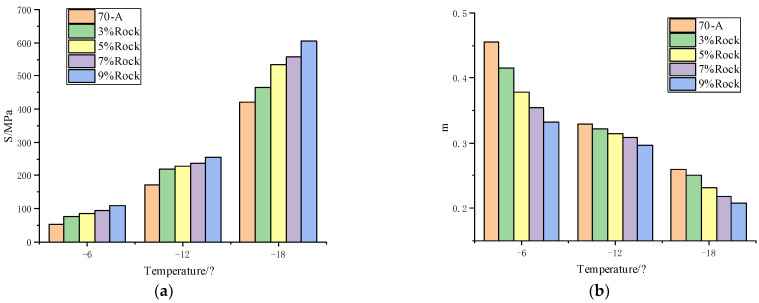
QRMA BBR results: (**a**) creep stiffness; (**b**) *m*-value.

**Figure 5 materials-15-07455-f005:**
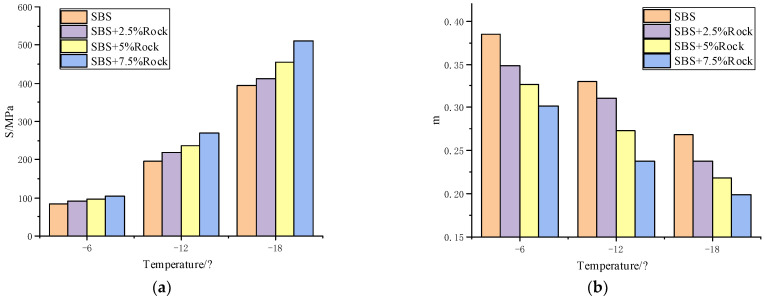
QRA-SBSMA BBR results: (**a**) creep stiffness; (**b**) *m*-value.

**Figure 6 materials-15-07455-f006:**
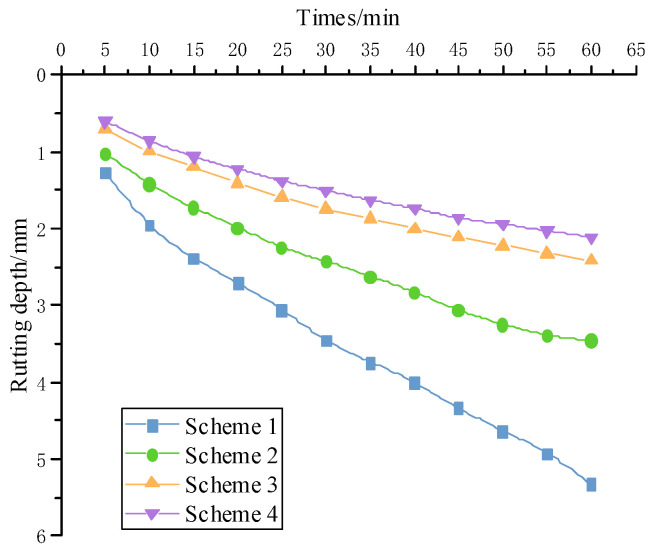
Rutting depth results over time.

**Figure 7 materials-15-07455-f007:**
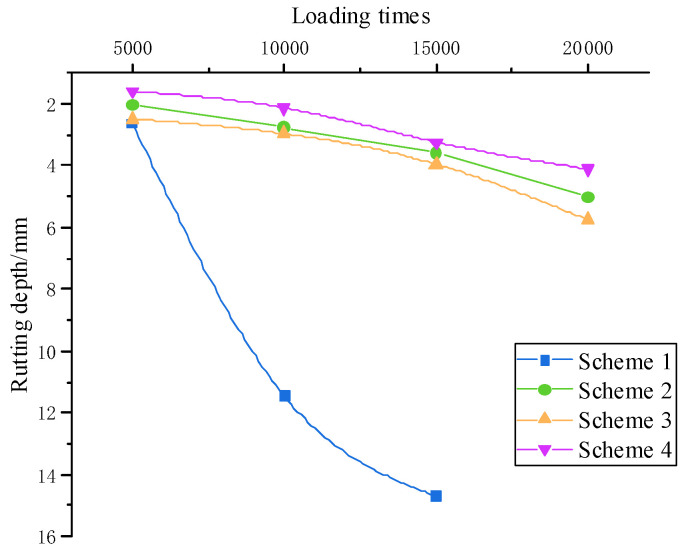
Comparison of rolling times and deformation of different types of mixtures.

**Figure 8 materials-15-07455-f008:**
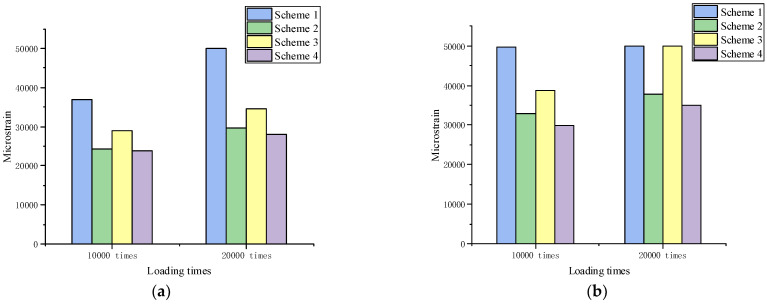
Creep testing results: (**a**) 30 °C; (**b**) 45 °C; (**c**) 60 °C; (**d**) Fn value.

**Figure 9 materials-15-07455-f009:**
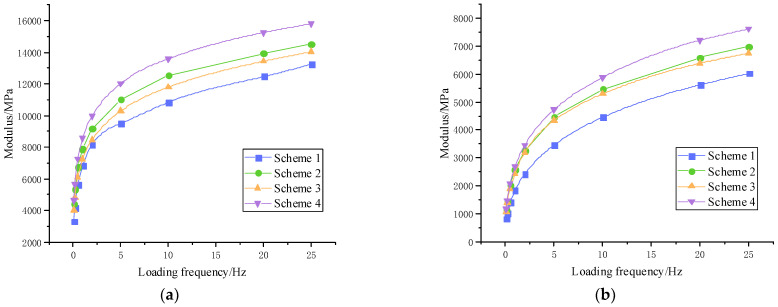
Modulus change results: (**a)** 30 °C, (**b**) 45 °C, (**c**) 55 °C, (**d**) 60 °C.

**Figure 10 materials-15-07455-f010:**
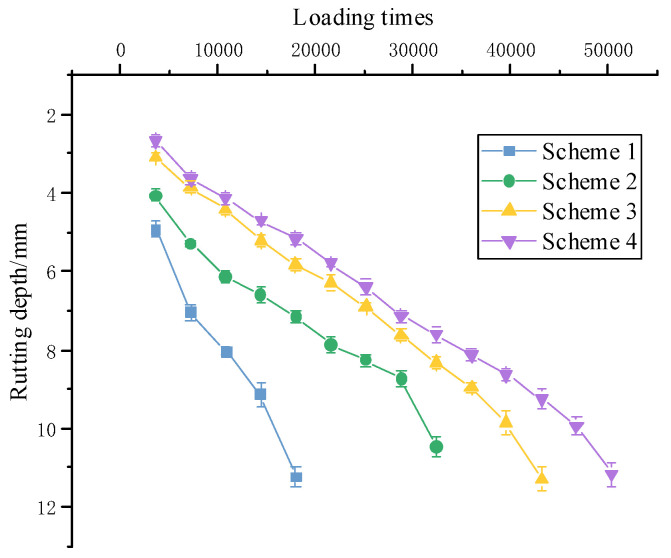
Accelerated loading testing rutting depth comparison.

**Figure 11 materials-15-07455-f011:**
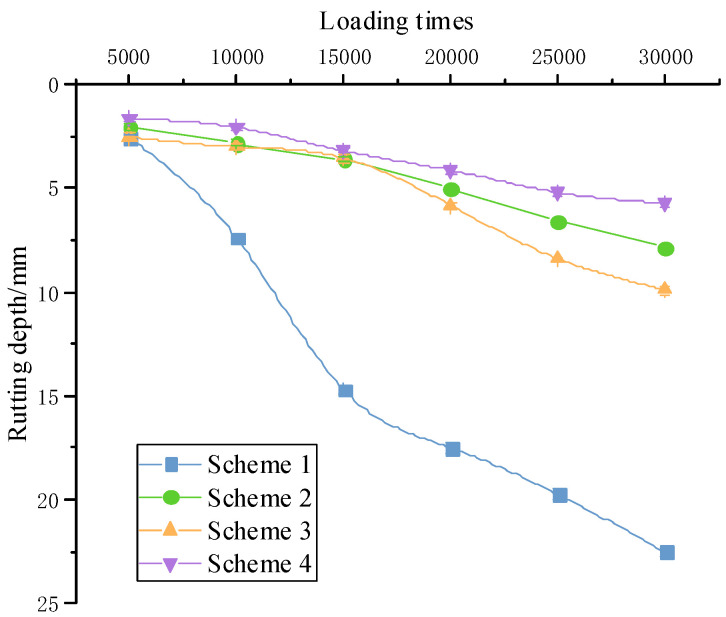
Rutting depth deformation pattern.

**Table 1 materials-15-07455-t001:** Test results of base asphalt performance index.

Index	Penetration/(0.1 mm)	Penetration Index	Softening Point/°C	Ductility/cm	RTFOT		
					Residual Penetration/%	Ductility/cm	Quality Change/%
Test results	69	0.56	49.5	140	75	14	−0.31
Technical requirement	60~80	−1.8~+1.0	≥45	≥100	≥58	≥4	−1.5~+1.0

**Table 2 materials-15-07455-t002:** Test results of Qingchuan rock asphalt performance index.

Index	Appearance	Asphalt Content/%	Ash Content/%	Water Content/%	Density/(g cm^−3^)
Test results	Black-brown powder	89.2	8.5	1.12	1.14
Technical requirement	Black, dark brown	/	≤10	≤2	1.10~1.20

**Table 3 materials-15-07455-t003:** Test results of SBS-modified asphalt performance index.

Index	Penetration/(0.1 mm)	Penetration Index	Softening Point/°C	Ductility/cm	RTFOT		
					Residual Penetration/%	Ductility/cm	Quality Change/%
Test results	46	0.35	72	28.8	79	19	−0.16
Technical requirement	40~60	≮0	≥60	≥20	≥65	≥15	−1.0~+1.0

**Table 4 materials-15-07455-t004:** Physical properties of limestone powder.

Index	Apparent Density	Water Content/%	Sieve Size		
			<0.6 mm	<0.15 mm	<0.075 mm
Test results	2.625	0.72	100	96.2	87.6
Technical requirement	≥2.5	≤1.0	100	90–100	75–100

**Table 5 materials-15-07455-t005:** Test results of Qingchuan rock-modified asphalt performance index.

Index	Penetration (25 °C, 100 g, 5 s)/(0.1 mm)	Softening Point/°C	Ductility (15 °C)/cm
70-A + 3%Rock	47	54	38
70-A + 5% Rock	45	56	33
70-A + 7% Rock	44	59	29
70-A + 9% Rock	42	61	24
Technical requirement	40~50	≥52	Actual results

**Table 6 materials-15-07455-t006:** Test results of Qingchuan rock–SBS-composite-modified asphalt performance index.

Index	Penetration (25 °C, 100 g, 5 s)/(0.1 mm)	Softening Point/°C	Ductility (10 °C)/cm
SBS + 2.5%Rock	34	91	46
SBS + 5%Rock	31	93	38
SBS + 7.5%Rock	27	94	30
Technical requirement	25~35	≥70	≥20

**Table 7 materials-15-07455-t007:** AC-10 and AC-16 mineral aggregate gradation.

Type	The Following Sieve Hole Pass Rate/mm										
	19	16	13.2	9.5	4.75	2.36	1.18	0.6	0.3	0.15	0.075
AC-10	100	100	100	99.2	62.4	38.2	26.9	18.1	12.8	10.2	6.5
AC-16	100	92.2	85.7	78.9	47.3	30.5	22.0	16.1	11.9	9.1	6.3

**Table 8 materials-15-07455-t008:** Design scheme of double-layer composite structure.

Scheme Design	Scheme 1	Scheme 2	Scheme 3	Scheme 4
Upper layer (AC-10)30 mm	70-A	7%Rock	SBS	SBS + 2.5%Rock
Lower layer (AC-16)50 mm	70-A	70-A	70-A	70-A

**Table 9 materials-15-07455-t009:** High-temperature PG classification results of QRMA.

Index	70-A	3% Rock	5% Rock	7% Rock	9% Rock
PG grade	64	70	70	76	76

**Table 10 materials-15-07455-t010:** High-temperature PG classification results of QRA-SBSMA.

Index	SBS	SBS + 2.5% Rock	SBS + 5% Rock	SBS + 7.5% Rock
PG grade	76	82	88	94

**Table 11 materials-15-07455-t011:** Low-temperature PG classification results of QRMA.

Index	70-A	3% Rock	5% Rock	7% Rock	9% Rock
PG grade	−16	−16	−16	−16	−10

**Table 12 materials-15-07455-t012:** Low-temperature PG classification results of QRA-SBSMA.

Index	SBS	SBS + 2.5% Rock	SBS + 5% Rock	SBS + 7.5% Rock
PG grade	−22	−22	−16	−16

**Table 13 materials-15-07455-t013:** Dynamic stability results of composite structure.

Scheme	Scheme 1	Scheme 2	Scheme 3	Scheme 4
Dynamic stability/mm	1058	1634	2210	2466

## Data Availability

Data available on request due to restrictions eg privacy or ethical. The data presented in this study are available on request from the corresponding author. The data are not publicly available due to the study data was not links to publicly archived datasets.

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
