# Peer review of "Research on High-Temperature Evaluation Indexes and Performance of Qingchuan Rock-SBS Composite Modified Asphalt"

_materials, 2022, doi:10.3390/ma15217455_

Round 1

Reviewer 1 Report

The authors reported mostly rheological aproaches with the temperature  and composition variations. The manuscript's topic is interesting due to its direct applicability in human lives. the manuscript is well-organized and the proposed outcome was achieved. there are minor changes which are proposed to be done:

1.Materials section. The authors presented somehow the materials to be analyzed, but no clear list or statement that they will vary the composition of asphalt; moreover the materials used for the modification were not clearly stated together with their provevience. Therefore, the authors are requested to reorganized better this section. 

 2. Rheological measurements  were mentioned a method, but no described; test conditions, parameters, devices, etc. are needed to be mentioned.

3. section 2.3. High temperature performance was described, but no range of temperature tests was inserted; the authors are requested to update

4. preparation method of the composite is included in the section of rheological study; a section with preparation of composite together with all details on the matrices should be included.

5. Rheological results: the authors mentioned a parameter, "The rutting factor" and it was not explained what information regarding the structure is giving and how can it be correlated with the classic dynamic moduli.

6. What is the highest temperature the prepared composite is resistant to? The optimal composition and temperature were not understood.

7. the authors mentioned about 4 schemes; what is the meaning of which and the differences and importance to study them

8. Conclusion: phrases need reformulation and English check e.g. "And we characterize the performance improvement effect of Qingchuan rock asphalt  on SBS modified asphalt to consummate the high-temperature evaluation system" not finished phrase. Furthermore, conclusion part should be completely reformulated as the authors did not finalize their expected outcome: till what temperature, other aspects on the composition range.

Author Response

Thank you for your kind suggestion. My revised manuscript is attached, please check it and improve your valuable comments. I will accept all the comments humbly and revise them one by one.

Reviewer 2 Report

This research assess the high-temperature performance of Qingchuan rock asphalt and SBS composite modified asphalt. high and low temperature performance of asphalt binder were examined and also the structural performance of asphalt was evaluated. The topic is of interest and the study is comprehensive. The approach and experimental program as well as data analysis are sound leading to reliable conclusions. The text is well written, but language editing can improve the quality and is recommended.

My main criticism towards this study is the lack of novelty. The usage of rock asphalt is widely used as you stated in the introduction and some studies showed "excellent high temperature performance and durability and also good low-temperature performance ". Therefore, I think authors must be able to justify the novelty of their research in a separate section like 'goals and objectives'. There you should also describe the research limitations and contribution to the body of the literature.

Author Response

(The authors gave the same response as above.)

Reviewer 3 Report

The article presents the results of tests of base asphalt and SBS asphalt in terms of modification with rock asphalt. The next step is to use them in the composition of asphalt mixtures.

The article presents interesting results. However, the results add nothing to civil engineering and transportation expertise. The use of rock asphalt in a modification is known and it is known to increase the temperature resistance.

Although the presented results are interesting, they require certain corrections and additions:

1. In Tables 1 and 3, the Ductility value is lower for SBS-Modified Asphalt. This is not normal. Please analyze these results.

2. In Fig. 3, the information should be completed which type of asphalt is presented in Fig. 3a and Fig. 3b.

3. Explain the descriptions in tab. 6 e.g. 70A; SBS + 2.5% Roc. The symbols are not in the article.

4. Test standards in sections 2.3.1 - 2.3.4 should be completed.

5. It is necessary to provide the number of repetitions to determine the average results that are presented on the charts. Please show "error bars" in the graphs.

6. The literature review must be expanded by authors who carry out research with other types of "rock asphalt".

7. Expand the description with "Qingchuan rock asphalt".

8. The article should be supplemented with the results of asphalt ductility and resistance to low temperatures.

Author Response

(The authors gave the same response as above.)

Reviewer 4 Report

The work is interesting but it looks to me that the background study was not investigated well. There are many studies in this area. In fact, the design of pavement using mechanistic method is the key research area at present. This study has some new finding. But it is impossible to understand the motivation and improvement with comparison to the existing knowledge. Please discuss the literature more elaborately using more examples. Most, if not all, of the references used are local to authors. Please use some international references. One example, pavement design – materials, analysis and highways, mcgraw hill.  The conclusion section can be rewritten to concisely give the conclusion. In current state, this section became a discussion section. Or, a new conclusion section can be added.

Author Response

(The authors gave the same response as above.)

Round 2

Reviewer 3 Report

Thank you for considering your comments.

1. The information on the temperature in the measurement of "Ductility" should be completed.

2. Error bars should be completed for all graphs.